# How accurately can one predict drug binding modes using AlphaFold models?

Masha Karelina[1,2,3,4,5], Joseph J Noh[2,3,4,5], Ron O Dror[1,2,3,4,5]*

[1]Biophysics Program, Stanford University, Stanford, United States; [2]Department of Computer Science, Stanford University, Stanford, United States; [3]Department of Molecular and Cellular Physiology, Stanford University School of Medicine, Stanford, United States; [4]Department of Structural Biology, Stanford University School of Medicine, Stanford, United States; [5]Institute for Computational and Mathematical Engineering, Stanford University, Stanford, United States

*For correspondence: ron.dror@stanford.edu

Competing interest: The authors declare that no competing interests exist.

**Abstract** Computational prediction of protein structure has been pursued intensely for decades, motivated largely by the goal of using structural models for drug discovery. Recently developed machine-learning methods such as AlphaFold 2 (AF2) have dramatically improved protein structure prediction, with reported accuracy approaching that of experimentally determined structures. To what extent do these advances translate to an ability to predict more accurately how drugs and drug candidates bind to their target proteins? Here, we carefully examine the utility of AF2 protein structure models for predicting binding poses of drug-like molecules at the largest class of drug targets, the G-protein-coupled receptors. We find that AF2 models capture binding pocket structures much more accurately than traditional homology models, with errors nearly as small as differences between structures of the same protein determined experimentally with different ligands bound. Strikingly, however, the accuracy of ligand-binding poses predicted by computational docking to AF2 models is not significantly higher than when docking to traditional homology models and is much lower than when docking to structures determined experimentally without these ligands bound. These results have important implications for all those who might use predicted protein structures for drug discovery.

## eLife assessment

This **important** study presents findings with broad implications for the use of AlphaFold 2 models in ligand binding pose modeling, a common task in protein structure modeling. The computational experiments and analyses provide **compelling** results for the GPCR protein family data, but the conclusions are likely to apply also to other proteins and they will therefore be of interest to biophysicists, physical chemists, structural biologists, and anyone interested or involved in structure-based ligand discovery.

## Introduction

Recent breakthroughs in machine learning have substantially improved the accuracy of protein structure prediction, to the point that some have declared the problem solved (*Baek et al., 2021*; *Jumper et al., 2021*; *Ourmazd et al., 2022*). Indeed, in the most recent round of the Community Assessment of Structure Prediction (CASP)—a blind protein structure prediction competition—AlphaFold 2 (AF2) demonstrated an unprecedented ability to predict protein structures with atomic accuracy, sometimes rivaling the accuracy of certain experimental methods for structure determination (*Kryshtafovych et al., 2021*; *Tejero et al., 2022*).

These advances have generated tremendous excitement, focused in particular on their potential impact on drug discovery (*Lowe, 2022*; *Thornton et al., 2021*; *Toews, 2021*). The vast majority of drug targets are proteins, and structures play a critical role in rational drug design (*Anderson, 2003*; *Kolb et al., 2009*). In particular, rational drug design strategies frequently require determining structures of the target protein bound to many ligands, as information on how these ligands bind is critical both to discovering early-stage drug candidates and to optimizing their properties (*Maveyraud and Mourey, 2020*; *Warren et al., 2012*). Determining these structures experimentally is typically slow and expensive, and sometimes proves impossible (*Slabinski et al., 2007*). Tremendous effort has thus gone into development of computational docking methods for predicting ligand-binding modes, which have dramatically accelerated this process in cases where a high-resolution experimentally determined structure of the target protein is available (*Ferreira et al., 2015*; *Pinzi and Rastelli, 2019*).

However, the metrics generally used to evaluate protein structure prediction methods—including all of those used in CASP—focus on the overall accuracy of predicted structures, rather than on their utility in predicting ligand binding. This has left the large community of academic and industrial researchers who wish to use protein structures to design drugs and other ligands unsure of whether, when, and how to use the new structural models. Improvements in protein structure prediction are widely assumed to lead to better models of ligand-binding pockets and thus more accurate predictions of ligand binding (*Beuming and Sherman, 2012*; *Bordogna et al., 2011*; *Erickson et al., 2004*), but to what extent do these assumptions hold?

Here, we address these questions by evaluating (1) the structural accuracy of ligand-binding pockets in modeled proteins and (2) the accuracy of ligand-binding poses predicted using these models, where a ligand's binding pose is specified by the 3D coordinates of its atoms when bound to the target protein. We evaluate models generated both by AF2 and by a traditional template-based modeling strategy, systematically comparing both to experimentally determined structures.

Several recent studies have experimented with the use of models generated by AF2—and related protein structure prediction approaches such as RoseTTAFold (*Baek et al., 2021*)—for tasks related to drug design, with mixed results (*Díaz-Rovira et al., 2023*; *He et al., 2023*; *Heo and Feig, 2022*; *Lee et al., 2022*; *Lee et al., 2023*; *Liang et al., 2022*; *Qiao et al., 2022*; *Scardino et al., 2023*; *Wong et al., 2022*). Our study stands apart in two regards. First, we use structural models generated without any use of known structures of the target protein. For machine-learning methods, this requires ensuring that no structure of the target protein was used to train the method. Second, we perform a systematic comparison that takes into account the variation between experimentally determined structures of the same protein when bound to different ligands.

Our results reveal both strengths and weaknesses of AF2 models. The structural accuracy of ligand-binding pockets in AF2 models is usually substantially higher than that of traditional homology models. Indeed, the typical difference between corresponding binding pockets in an AF2 model and in an experimentally determined structure is only slightly larger than the typical difference between two experimentally determined structures of the same protein with different ligands bound. Surprisingly, however, ligand-binding poses predicted given AF2 models are not significantly more accurate than those predicted given traditional models and are much less accurate than those predicted by docking computationally to experimentally determined protein structures. Our results provide guidelines as to how AF2 models should—and should not—be used for effective ligand-binding prediction. Our findings also suggest opportunities for improving structure prediction methods to maximize their impact on drug discovery.

## Results
### Selection of proteins and models to ensure a fair comparison

Evaluating the accuracy of predicted ligand-binding poses requires that we examine protein–ligand complexes whose structures have been determined experimentally. We wish to ensure, however, that the predicted structures of a given protein are not informed by experimentally determined structures of that same protein.

AF2 and other recent neural-network-based protein structure prediction methods use experimentally determined protein structures in two ways: first, to train the neural network and second, as

templates for prediction of individual protein structures (*Baek et al., 2021*; *Jumper et al., 2021*). The AF2 neural network was trained on structures in the Protein Data Bank (PDB) (*Berman et al., 2000*) as of April 30, 2018 so we evaluated structural models of proteins for which no experimentally determined structure was available as of that date. In addition, when generating models using AF2, we used only structures that were available on that date as templates (see Methods). For a fair comparison, we also ensured that the traditional template-based models we used were not informed by any structures that became available after that date. In particular, we used traditional models that were published before April 30, 2018.

We selected all proteins, ligands, and structural models used in this study in a systematic manner (see Methods). We focused on one class of drug targets—G-protein-coupled receptors (GPCRs)—for three reasons. First, GPCRs represent the largest class of drug targets by a substantial margin, with 35% of FDA-approved drugs targeting a GPCR (*Sriram and Insel, 2018*). Second, GCPR-binding pockets are very diverse, reflecting the fact that GPCRs have evolved to bind very different ligands with high specificity (*Venkatakrishnan et al., 2013*). Third, the GPCRdb project maintains a historical database of template-based models of GPCRs (*Pándy-Szekeres et al., 2018*). By selecting the template-based models listed in GPCRdb in April 2018, we ensured that these models were not informed by any structures that became available after April 30, 2018.

## Binding pocket structures are much more accurate in AF2 models than traditional models

Because proteins are flexible, each protein can adopt multiple structures. In particular, the structure of a protein—especially the structure of the binding pocket—depends on which ligand is bound. To quantify this natural variation, we compared all pairs of experimentally determined structures of the same protein to one another, using root mean squared deviation (RMSD) as a metric (see Methods).

AF2 does not allow one to specify a bound ligand when generating a protein model, and the template-based models listed in GPCRdb were also generated without specification of a bound ligand. We thus evaluated the accuracy of each model by computing its RMSD to that of all available experimental structures of the same protein. We compared the resulting distribution of RMSDs to the distribution of RMSDs between structures of the same protein with different ligands bound.

As expected based on previously reported results for other proteins (*Jumper et al., 2021*), we find that the global structural accuracy of the AF2 models is substantially better than that of the traditional template-based models. In particular, when computing RMSDs based on all non-hydrogen atoms in each protein, we find a median RMSD of 2.9 Å between AF2 models and corresponding experimentally determined structures, compared to a median RMSD of 4.3 Å for traditional models (*Figure 1— figure supplement 1*).

Next, we turn our attention to the orthosteric binding pocket of each protein—that is, the region where native ligands and most other ligands, including those in our dataset, bind. AF2 was trained to optimize global structural accuracy, with an emphasis on correctly predicting the structure of the protein backbone. The accuracy of binding pocket prediction, on the other hand depends heavily on side-chain conformations in this local region. Nevertheless, we find that AF2 predictions of binding pocket structures are typically very accurate.

In fact, the binding pocket RMSD between an AF2 model and an experimentally determined structure is typically nearly as low as the RMSD between two experimentally determined structures of the same protein with different ligands bound (*Figure 1*). In contrast, binding pockets in the traditional template-based models are much less accurate, with a median RMSD of 3.3 Å for traditional models vs. 1.3 Å for AF2 models. We note that the binding pockets of a few AF2 models are highly inaccurate (approximately 5 Å RMSD), as is the case for several of the traditional models.

## Binding pose prediction using AF2 models or traditional models yields similar accuracy

Next, we assess the accuracy of ligand-binding poses predicted by computational docking to AF2 models and traditional template-based models. For each ligand we consider, the binding pose to a target protein is known based on an experimentally determined structure of the ligand bound to the protein, which we call the *reference structure*. Using industry-standard software, we docked each ligand to both AF2 and GPCRdb models of the protein. A predicted pose is considered correct if

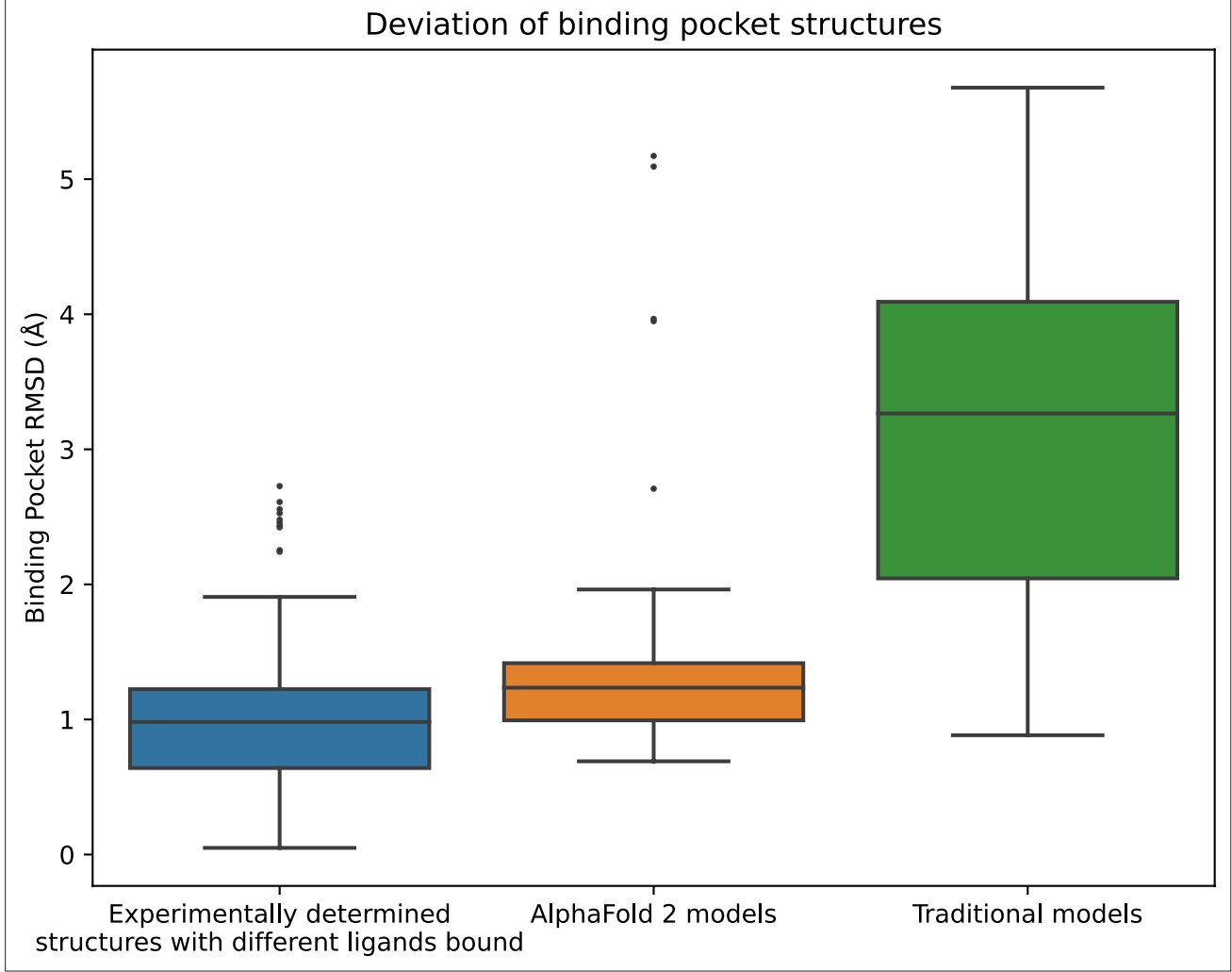

**Figure 1.** Structural accuracy of modeled binding pockets. Binding pockets are defined to include all amino acid residues with any atom within 5 Å of the ligand in an experimentally determined structure. We compute all-atom binding pocket root mean squared deviations (RMSDs) between each modeled structure and all experimentally determined structures of the same protein. For comparison, we also compute binding pocket RMSDs between all pairs of experimentally determined structures of the same protein with different ligands bound. The middle line of each box in the plot is the median RMSD, with the box extending from the first to the third quartile and defining the 'interquartile range'. Whiskers extend to last data points that are within 150% of the interquartile range, and outlier data points beyond those are shown individually. The plotted data is based on 150 RMSD values (comparisons) for experimentally determined structures with different ligands bound, 52 for AlphaFold 2 models, and 78 for traditional models.

The online version of this article includes the following figure supplement(s) for figure 1:

**Figure supplement 1.** Structural accuracy of modeled proteins.

**Figure supplement 2.** Structural accuracy of modeled binding pockets.

its RMSD from the experimentally determined pose is ≤2.0 Å (**Figure 2**), a widely used criterion for correct pose prediction (**Erickson et al., 2004**; **Gohlke et al., 2000**; **Paggi et al., 2021**).

For comparison, we also dock each ligand to structures of the same protein determined experimentally with other ligands bound (cross-docking). Note that when calculating pose prediction accuracy (**Figure 2**), we exclude cases where the ligand is docked back to the reference structure; such 'self-docking' yields much higher accuracy (**Figure 2—figure supplement 1**) but is of little practical utility, because in practice one wishes to predict binding poses that have not already been determined experimentally.

Strikingly, binding pose prediction accuracy was similar when using AF2 models or traditional template-based models, despite the fact that binding pocket structures were substantially more accurate in AF2 models. Although the fraction of ligands docked correctly was slightly higher when using

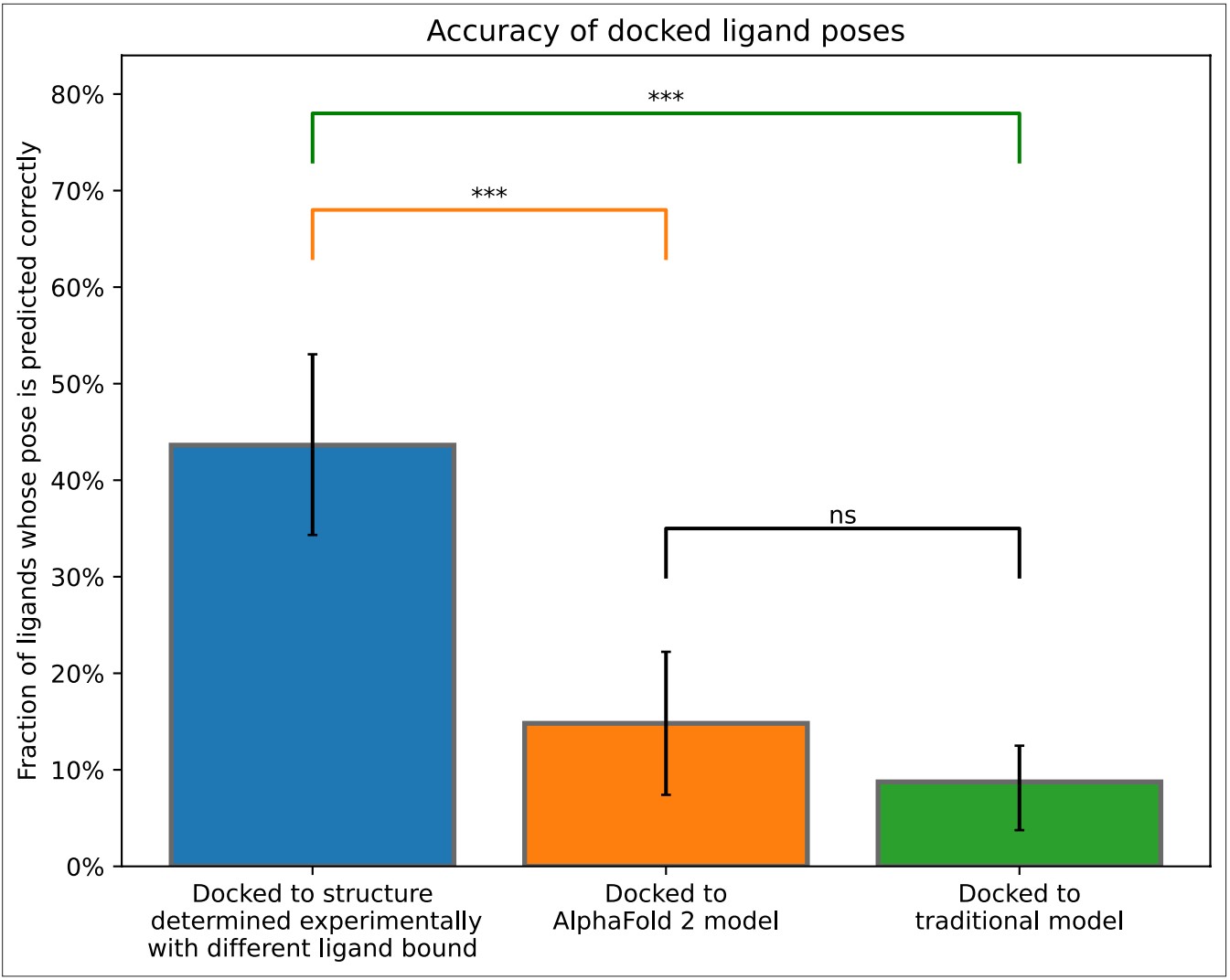

**Figure 2.** Accuracy of ligand-binding poses predicted by computational docking to AlphaFold 2 models, traditional template-based models, or protein structures determined experimentally in complex with a ligand different from the one being docked. We plot the fraction of docked ligands whose pose is predicted correctly (see Methods). Error bars are 90% confidence intervals calculated via bootstrapping. *** for p-values <0.001, ns for p-values >0.05. 40 - 54 ligands were evaluated for each set of structures or models.

The online version of this article includes the following figure supplement(s) for figure 2:

**Figure supplement 1.** Accuracy of ligand-binding poses predicted by computational docking to AlphaFold 2 models, traditional template-based models, or experimentally determined protein structures.

**Figure supplement 2.** Accuracy of ligand-binding poses predicted by computational docking to AlphaFold 2 models, traditional template-based models, or protein structures determined experimentally in complex with ligands different from the one being docked or very different from the one being docked.

AF2 models (15%) than when using traditional models (9%), this difference was not statistically significant ($p > 0.5$). On the other hand, binding pose prediction accuracy was far higher when docking to experimentally determined protein structures (44%), even though these structures were determined in complex with ligands different from those being docked.

The docked poses analyzed above were determined with the widely used commercial docking software package Glide, run in SP mode (for 'standard precision') (*Friesner et al., 2004*). To verify the robustness of our results, we repeated all docking experiments using a completely different docking software package—Rosetta docking (*Park et al., 2021*)—as well as with Glide run in XP mode (for 'extra precision') (*Friesner et al., 2006*). In both cases, we again found that, when used for ligand-binding pose prediction, AF2 models substantially underperformed experimentally determined

structures and did not significantly outperform traditional models (*Figure 2—figure supplement 2*). In fact, the difference in binding pose prediction accuracy between AF2 models and traditional models was even smaller when using Rosetta docking or Glide XP than when using Glide SP.

## Closer look at discrepancy between structural accuracy and pose prediction accuracy

The results of the previous two sections contrast sharply with one another. AF2 models nearly match experimentally determined structures—and are much better than traditional template-based models—in binding pocket accuracy, as measured by a standard structural metric. Yet AF2 models fare much worse than experimentally determined structures—do not significantly outperform traditional models—when used to predict ligand-binding poses. *Figures 3 and 4* provide illustrative examples.

To further quantify these results, we compare pose prediction accuracy for structures and models with similar binding pocket accuracy. In particular, we calculate pose prediction accuracy as a function of binding pocket accuracy (RMSD) for both experimentally determined structures and models. As expected, we find that average pose prediction accuracy increases as binding pocket accuracy increases (i.e., RMSD to reference structure decreases). This trend holds for experimentally determined structures, AF2 models, and traditional template-based models (*Figure 5* and *Figure 5—figure supplement 1*). Yet for the same binding pocket accuracy (RMSD from the reference structure), AF2 models yield lower average pose prediction accuracy than experimentally determined structures (*Figure 3*). In contrast, in cases where traditional models and experimentally determined structures have similar binding pocket accuracies, they yield similar pose prediction accuracies (*Figure 5—figure supplement 1*).

We also considered the possibility that the improved performance of experimentally determined structures relative to AF2 models might be because, in some cases, the structure used for docking was determined with a bound ligand similar (though not identical) to the ligand being docked. Eliminating such cases from consideration does lead to decreases in both aggregate binding pocket accuracy and pose prediction accuracy of experimentally determined structures, but experimentally determined structures continue to yield significantly more accurate pose predictions than AF2 models ($p < 0.01$) (*Figure 1—figure supplement 1* and *Figure 2—figure supplements 1 and 2*).

## Discussion

Several recent papers have concluded that AF2 models fall short of experimentally determined structures when used for ligand docking (*Díaz-Rovira et al., 2023*; *He et al., 2023*; *Heo and Feig, 2022*; *Scardino et al., 2023*). Our results agree with this conclusion but go beyond it in several ways. We compare AF2 models and traditional template-based models, rigorously ensuring that none of these models was informed by experimentally determined structures of the protein being modeled. We find that, in terms of structural accuracy of the binding pocket, AF2 models are much better than traditional models; indeed, the RMSD between an AF2 model and an experimentally determined structure is typically comparable to the RMSD between two structures determined experimentally with different ligands bound. Yet, when used for docking, AF2 models yield pose prediction accuracy similar to that of traditional models and much worse than that of experimentally determined structures.

A few caveats are in order. First, building a traditional template-based model is not always possible. We only considered template-based models in cases where a structure was available for another protein with at least 40% sequence identity. However, 80% of human drug targets have at least 50% sequence identity to a protein whose structure was known in 2017 (*Somody et al., 2017*), and that number is even higher today. In addition, we note that the traditional models we used were generated relatively quickly by an academic group along with models for hundreds of other proteins. An expert working on a drug discovery project could frequently build a better template-based model—for example, by examining the predictive utility of several models (*Carlsson et al., 2011*; *Haddad et al., 2020*).

Second, our study examined only GPCRs. The results might in principle be different for other classes of drug targets. However, GPCRs have very diverse binding pockets, which have evolved to binding an extremely broad range of ligands, ranging from very small molecules to peptides and proteins, with high specificity (*Venkatakrishnan et al., 2013*). Our conclusions thus likely apply to many non-GPCR

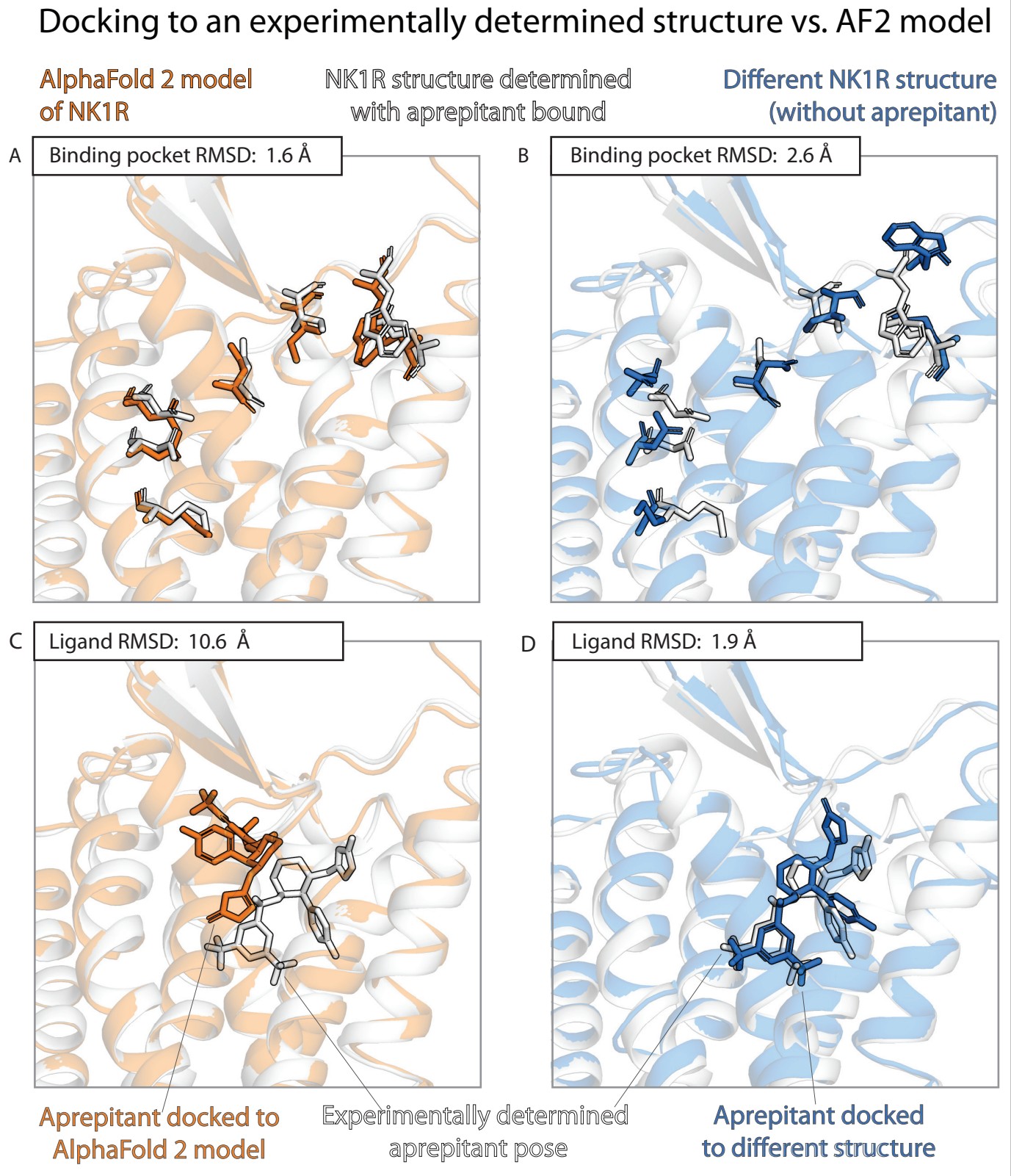

# Docking to an experimentally determined structure vs. AF2 model

**AlphaFold 2 model of NK1R**

NK1R structure determined with aprepitant bound

**Different NK1R structure (without aprepitant)**

A  Binding pocket RMSD: 1.6 Å

B  Binding pocket RMSD: 2.6 Å

C  Ligand RMSD: 10.6 Å

D  Ligand RMSD: 1.9 Å

**Aprepitant docked to AlphaFold 2 model**

Experimentally determined aprepitant pose

**Aprepitant docked to different structure**

**Figure 3.** An example in which docking to an AlphaFold 2 (AF2) model yields poor results even though the model's binding pocket has high structural accuracy. We predict the binding pose of the drug aprepitant to its target, the neurokinin-1 receptor (NK1R) given either the AF2 model (orange) of NK1R or the experimentally determined structure (blue, PDB entry 6E59) of NK1R bound to a different ligand, L760735. (**A, B**) The binding pocket of the AF2 model is more similar (lower root mean squared deviation [RMSD]) than the binding pocket of the L760735-bound structure to the binding pocket

*Figure 3 continued on next page*

*Figure 3 continued*

of the aprepitant-bound structure (the 'reference structure', white, PDB entry 6J20). Amino acid residues whose positions differ most from the reference structure are shown in sticks (see Methods). (**C, D**) The aprepitant binding pose predicted by docking is much less accurate (higher RMSD) when using the AF2 model than when using the L760735-bound structure. Ligand L76035 shares a scaffold with aprepitant; for completeness, we include another example with highly dissimilar ligands in *Figure 3—figure supplement 1*. We note that the experimentally determined L760735-bound structure is a low-resolution structure with suboptimal goodness of fit to experimental data; despite this, docking aprepipant to this structure yields an accurate pose.

The online version of this article includes the following figure supplement(s) for figure 3:

**Figure supplement 1.** An example in which docking to an AlphaFold 2 (AF2) model yields poor results even though the model's binding pocket has high structural accuracy.

drug targets—and GPCR targets alone account for roughly half of drug discovery projects. We also note that, for GPCRs specifically, one can often improve docking performance by using a model or an experimentally determined structure in the activation state to which a given ligand preferentially binds (*Heo and Feig, 2022*; *Lee et al., 2022*). In this study, we did not limit the ligands docked to each structure or model to those that preferentially bind one specific activation state. We made this choice for several reasons: many other classes of drug targets do not have well-defined activation states; in practical applications, the preferred activation state for a given ligand may not be known when one docks it; and AF2 does not produce models in specific activation states by default.

Third, we have not examined performance of the full range of available computational docking methods (*Pagadala et al., 2017*). Our results are, however, consistent across the three docking methods we examined: Glide SP, Glide XP, and Rosetta docking. In each case, AF2 models yield docking accuracy similar to that of traditional models and substantially worse than that of experimentally determined structures.

Our results have important implications for practitioners of structure-based drug discovery. Our findings suggest that, despite dramatic recent improvements in protein structure prediction, structural models (in particular, AF2 models) yield substantially poorer accuracy than experimentally determined structures in prediction of ligand-binding modes. Moreover, when a reasonable structural template is available—as is typically the case—AF2 models should not be expected to yield significantly better pose prediction accuracy than traditional models. Indeed, with sufficient effort, an expert may well be able to build a traditional template-based model that yields better performance than an AF2 model.

Despite these findings, we believe that AF2 and other recently developed protein structure prediction methods will prove valuable in certain structure-based drug discovery efforts. Up-to-date AF2 models for any human protein are readily available for download (*Tunyasuvunakool et al., 2021*; *Varadi et al., 2022*). In contrast, expertise is required to build a decent template-based model, and up-to-date, downloadable repositories of such models are not available for all drug targets. Also, in the relatively rare cases when no structural template is available for a drug target, approaches like AF2 may be the only realistic option short of solving a structure experimentally.

Our results also point to opportunities for developing improved computational modeling methods. Existing deep-learning methods for protein structure prediction were developed with the aim of maximizing structural accuracy for models. We find that the RMSD of a model—a classic structural accuracy metric—is not a good predictor of the accuracy that model yields for pose prediction. Future protein structure prediction methods might instead be designed to maximize utility for ligand-binding prediction, while still exploiting the deep-learning advances of AF2 and related recent methods. We also note that the vast majority of test cases used in the design of currently available docking software involved experimentally determined protein structures. One might be able to redesign docking methods to yield better performance given computationally predicted models.

## Methods
### Protein and ligand selection

We selected structures from the aggregated list of all experimentally determined human GPCR structures from GPCRdb (*Isberg et al., 2014*). We removed all proteins for which a structure was published in the PDB before April 30, 2018, including two Class C GPCRs for which structures of the extracellular domain (to which ligands bind) had been published before that date. We also removed proteins for which structures were available in complex with fewer than two unique orthosteric ligands. This left

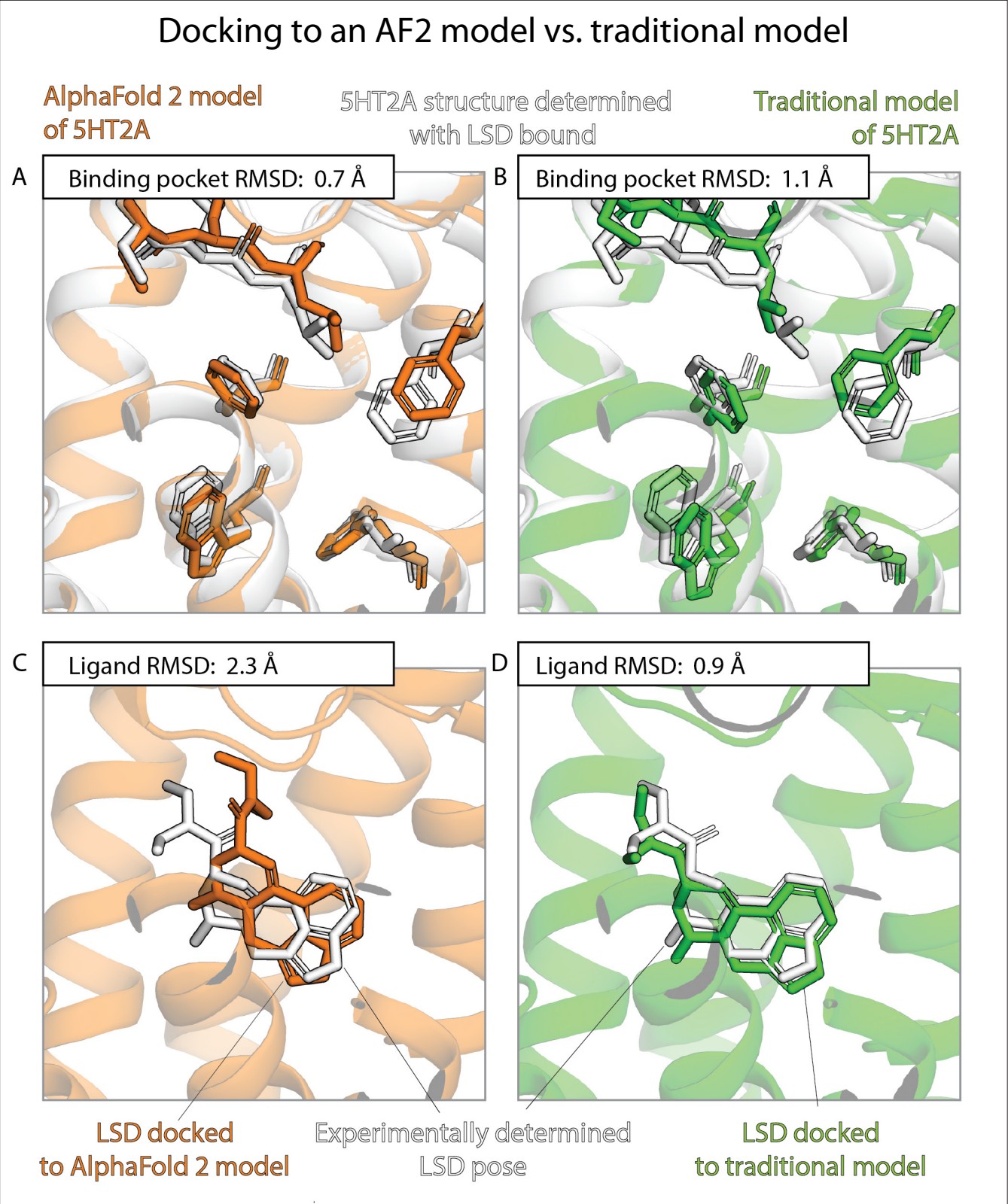

## Docking to an AF2 model vs. traditional model

AlphaFold 2 model of 5HT2A

5HT2A structure determined with LSD bound

Traditional model of 5HT2A

**A** Binding pocket RMSD: 0.7 Å

**B** Binding pocket RMSD: 1.1 Å

**C** Ligand RMSD: 2.3 Å

**D** Ligand RMSD: 0.9 Å

LSD docked to AlphaFold 2 model

Experimentally determined LSD pose

LSD docked to traditional model

**Figure 4.** An example in which docking to a traditional template-based model yields better results than docking to an AlphaFold 2 (AF) model, even though the AF2 model's binding pocket has higher structural accuracy. We predict the binding pose of the psychedelic LSD to its primary target, the serotonin 2A receptor (5HT2A) given either the AF2 model (orange) or a traditional model (green) of 5HT2A. (**A, B**) The binding pocket of the AF2 model is more similar (lower root mean squared deviation [RMSD]) than that of the traditional model to the binding pocket of the experimentally

*Figure 4 continued on next page*

*Figure 4 continued*

determined LSD-bound structure (the 'reference structure', white, PDB entry 6WGT). Amino acid residues that differ most from the reference structure are shown in sticks. (**C, D**) The LSD-binding pose predicted by docking is less accurate (higher RMSD) when using the AF2 model than when using the traditional model.

us with 54 experimentally determined structures of 18 GPCRs (17 class A, 1 class B). The full list of proteins and structures is in *Supplementary file 1*.

## Obtaining structural models
### AlphaFold 2
For each protein, we took the full protein sequence from Uniprot (*UniProt Consortium, 2023*) to generate five AF2 models, as was done for CASP (*Jumper et al., 2021*), and picked the top-scoring prediction as our model. We set the template cutoff date to 2018-04-30.

### GPCRdb
We downloaded template-based models from the GPCRdb archive dated 2018-04-06. We used all the inactive- and active-state models available, and averaged results as described in Statistical tools. We

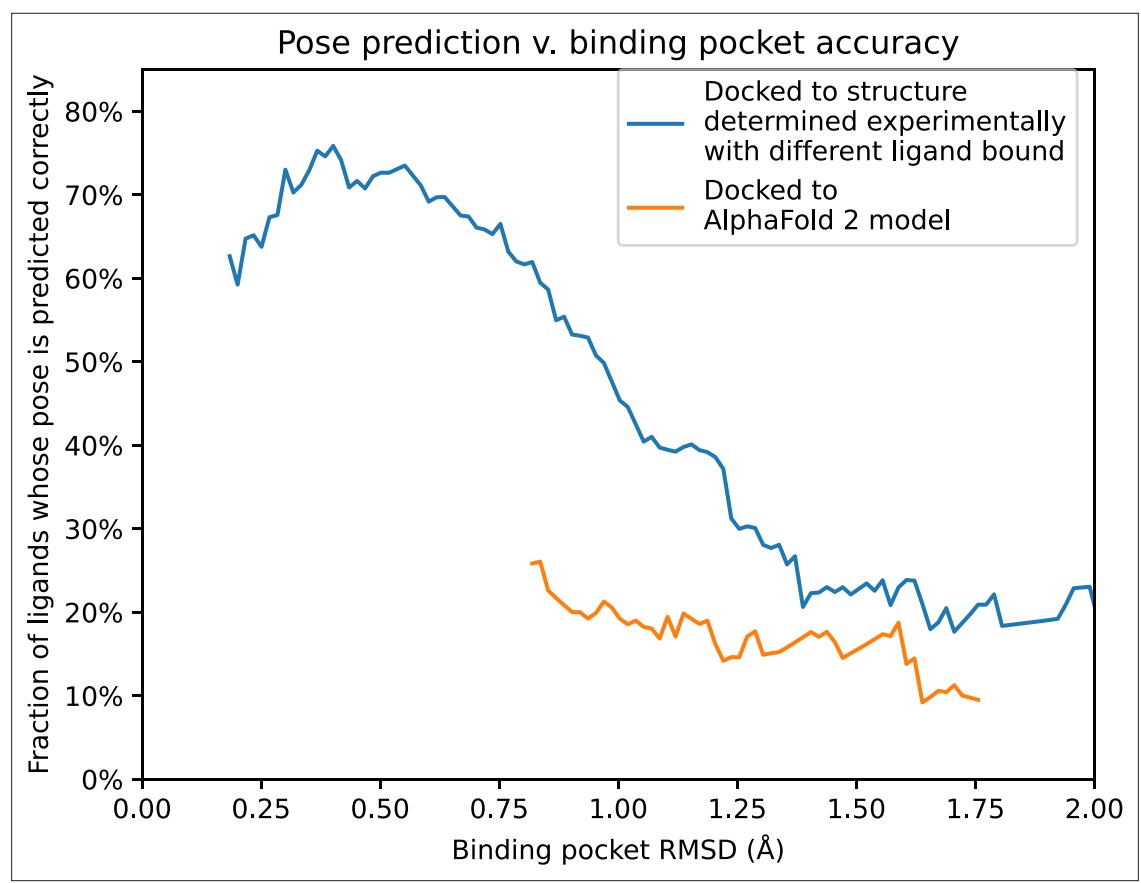

**Figure 5.** Pose prediction accuracy as a function of binding pocket structural accuracy when docking to AlphaFold 2 (AF2) models or experimentally determined structures. Docking to an experimentally determined structure generally leads to more accurate pose prediction than docking to an AF2 model with the same binding pocket root mean squared deviation (RMSD). The difference between the two curves is statistically significant for all binding pocket RMSD values below 1.1 Å (see Methods). See *Figure 5—figure supplement 1* for additional data, including results for traditional models and for various docking methods.

The online version of this article includes the following figure supplement(s) for figure 5:

**Figure supplement 1.** Pose prediction accuracy as a function of binding pocket structural accuracy when docking to AlphaFold 2 (AF2) models, experimentally determined structures or traditional template-based models, with different docking protocols.

excluded models for which the minimum sequence identity of the templates was below 40%, as such template-based models are typically not used for drug-binding predictions.

This led to exclusion of models for four GPCRs (AGRG3, TA2R, PE2R2, and PE2R4). GPCRdb did not include models for two of the GPCRs (GPBAR and PE2R3).

## Ligand preparation

Ligands were extracted from PDB structures using the Schrödinger Python API and then manually inspected to make sure we had chosen the ligand at the orthosteric site. With the same API, ligands were converted to SMILES strings. Ligands were then prepared with Schrödinger's LigPrep with default command line parameters (*Schrödinger LLC, 2021*).

Ligand similarity was defined as a ratio of the size of the maximum common substructure to the size of the smaller molecule. Ligand pairs for which this ratio was less than 0.5 were deemed very different ligands.

## Protein preparation

Experimentally determined structures were downloaded from the PDB. Only a single chain containing the ligand was kept, and all waters were removed. All structures and models were prepared with Schrödinger Protein Preparation Wizard (*Schrödinger LLC, 2021* ) following the same protocols as *Paggi et al., 2021*, including energy minimization. All the experimentally determined structures were determined with a ligand bound, and we retained this ligand during minimization. Computationally predicted models did not include a ligand.

## Glide docking

For Glide XP and Glide SP docking, we follow the docking protocol described in *Paggi et al., 2021*. The grid is centered at the geometric centroid of the ligand and is defined with an inner box with 15 Å sides and outer box with 30 Å sides. For predicted structural models the center of the box is determined by aligning the model to an experimentally determined structure of that protein and using the centroid of its ligand. For this alignment, we pick the structure with the alphabetically first PDB entry ID.

## Rosetta docking

For Rosetta docking, we used the GALigandDock protocol (*Park et al., 2021*). To prepare the proteins, PyMol (*Schrödinger, 2015*) was used to remove heteroatoms and all alternative locations for each atom. CONECT information was included. UCSF Chimera v1.16 (*Pettersen et al., 2004*) was used to initially remove all hydrogens, followed by their Dock Prep software to prepare the structure for docking.

For ligands, Open Babel v2.4.0 (*O'Boyle et al., 2011*) was used to generate 3D structures from ligand SMILES and add hydrogens at a pH of 7 (*O'Boyle et al., 2011*). Ambertools v22.0 (*Case et al., 2023*) was used to add AM1-BCC partial charges. The ligand was translated such that the center of mass of the ligand was positioned at center of the binding pocket of the prepared structure. The center of the binding pocket was defined as the center of mass of the ligand in the experimentally determined structure. Models were aligned to the experimentally determined structure of the protein which had the alphabetically first PDB entry ID. This structure's ligand center of mass was used to define the model's binding pocket.

## Structural comparisons

To calculate ligand pose RMSD, structures were first aligned on amino acid residues within 15 Å of bound orthosteric ligands, using Schrödinger's structalign tool (*Schrödinger LLC, 2021*). This alignment was used to calculate the RMSD of each docked ligand pose from the reference ligand pose.

Binding pocket RMSDs were calculated with a PyMOL script, considering all residues that are within 5 Å of the ligand in the reference structure (for both alignment and RMSD calculation). We included all non-hydrogen atoms in this calculation, except that for the backbone-only binding pocket RMSDs shown in *Figure 5—figure supplement 1* we included only backbone atoms.

We computed full-structure RMSDs with a similar PyMOL script, but taking into account all non-hydrogen atoms in the protein.

In *Figure 3*, *Figure 4*, and *Figure 3—figure supplement 1*, we highlight residues in the binding pocket whose positions differ most from the reference structure. To identify these residues, we aligned to the reference structure the structure and model (or the two models) being docked to. For each structure/model, we computed the RMSD for each residue within 5 Å of the ligand, and then for each residue determined the maximum RMSD across the two structures/models. We selected the residues with the largest maximum RMSDs.

## Statistical tools

p values were computed with two sided paired *t*-tests, using SciPy (*Virtanen et al., 2020*). Bootstrapping 90% confidence intervals were computed with the default 9999 resamples, also using SciPy.

In *Figure 2*, the docking accuracy is calculated as a weighted average across all docking results, with weights chosen such that each ligand–protein pair is represented equally. In particular, when multiple traditional template-based models of a single protein were available, we docked each ligand to each model but set the averaging weight to the inverse of the number of models docked to. Likewise, when multiple experimentally determined structures of a single protein were available for docking of a particular ligand (i.e., structures determined experimentally without that particular ligand bound), we docked the ligand to each structure but set the averaging weight to the inverse of the number of structures docked to.

In *Figure 5* and *Figure 5—figure supplement 1*, instead of simply binning structures and models in different RMSD ranges, we performed a smoothing analysis that does not impose arbitrary cutoffs between bins. The smoothed functions were created by running kernel smoothing over docking outcomes. The smoothing was done over data such that each data point was the binding pocket RMSD and the binary value indicating whether or not the predicted pose was correct. An Epanechnikov kernel (*Hastie et al., 2009*) was used, with a width of 0.25 Å. To avoid inaccurate boundary condition behaviors, each end the kernel was only evaluated when the center of the kernel overlapped with the first or last data point on the *X*-axis. To calculate statistical significance and p values, a bootstrapping significance test was conducted for each interval on the *X*-axis. Additionally, in *Figure 5—figure supplement 1*, we show 90% confidence intervals that are also computed with bootstrapping.

## Acknowledgements

We thank Joseph Paggi, John Wang, Akshat Nigam, and all members of the Dror lab for assistance and insightful comments. This work was supported by Novo Nordisk and an NSF Graduate Research Fellowship to MK.

## Additional information

### Funding

| Funder | Grant reference number | Author |
|---|---|---|
| National Science Foundation | Graduate Research Fellowship Program | Masha Karelina |
| Novo Nordisk | | Ron O Dror |

The funders had no role in study design, data collection, and interpretation, or the decision to submit the work for publication.

### Author contributions

Masha Karelina, Conceptualization, Data curation, Investigation, Writing – original draft, Writing – review and editing; Joseph J Noh, Investigation; Ron O Dror, Conceptualization, Supervision, Writing – original draft, Writing – review and editing

### Author ORCIDs

Masha Karelina ⓘ http://orcid.org/0000-0003-1880-4536
Ron O Dror ⓘ https://orcid.org/0000-0002-6418-2793

Reviewer #1 (Public Review): https://doi.org/10.7554/eLife.89386.2.sa1
Reviewer #2 (Public Review): https://doi.org/10.7554/eLife.89386.2.sa2
Author Response https://doi.org/10.7554/eLife.89386.2.sa3

## Additional files

### Supplementary files
- Supplementary file 1. The full list of proteins and structures.
- MDAR checklist

### Data availability
Data and analysis scripts are supplied in https://doi.org/10.5281/zenodo.10069998.

The following dataset was generated:

| Author(s) | Year | Dataset title | Dataset URL | Database and Identifier |
|---|---|---|---|---|
| Karelina M, Noh J, Dror R | 2023 | Data and structures for "How accurately can we predict binding poses with AlphaFold models? | https://zenodo.org/records/10069998 | Zenodo, 10.5281/zenodo.10069998 |

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

## Appendix 1

**Appendix 1—key resources table**

| Reagent type (species) or resource | Designation | Source or reference | Identifiers | Additional information |
|---|---|---|---|---|
| Software | Schrödinger tools | Schrödinger | Version: 2021-1 glide-v9.0 | Glide, Maestro, Protein Preparation Wizard |
| Software | PyMOL | Schrödinger | PyMOL 3.8 | |
| Software | AlphaFold 2.0.1 | https://doi.org/10.1038/s41586-021-03819-2 | | |
| Software | Rosetta GALigandDock | https://doi.org/10.1021/acs.jctc.0c01184 | | |

