## [Editor Report · eLife assessment]

This **important** study presents findings with broad implications for the use of AlphaFold 2 models in ligand binding pose modeling, a common task in protein structure modeling. The computational experiments and analyses provide **compelling** results for the GPCR protein family data, but the conclusions are likely to apply also to other proteins and they will therefore be of interest to biophysicists, physical chemists, structural biologists, and anyone interested or involved in structure-based ligand discovery.

---

## [Referee Report · Reviewer #1 (Public Review)]

The authors assess the accuracy of AlphaFold2 (AF2) structures for small molecule ligand pose prediction versus the accuracy with traditional template-based homology models and experimental crystal structures (with a different ligand). They take a careful, rigorous approach leveraging the wealth of structural data around the GPCR protein family and using state-of-the-art docking methods. They find that binding sites are significantly more accurately modeled by AF2 compared to traditional template-based approaches, but this does not translate to greater accuracy in small-molecule docking pose prediction. The important findings around this conclusion have broad implications for the use of AF2 models in ligand binding pose prediction for proteins and drug design.

Strengths:

The strength of the work is the rigor and careful, thoughtful comparison that cleverly leverages the cut-off date of April 30th, 2018 used in the training of AF2. While the authors list their limited number of docking methods as a caveat, the fact that they use three state-of-the-art ligand docking methods is a strength of the work; many studies use just one. The rigorous analysis of the binding site RMSD and docked ligand pose RMSD is novel to my knowledge and is particularly insightful.

Weaknesses:

The authors are rigorous in their approach by using state-of-the-art workflows that are high-throughput in nature. However, human expert-refined models and expert selection from multiple models could improve the results of ligand pose prediction when using protein models. The authors alluded to this for traditional models but this can also be true when starting from AF2 models. This is difficult to test systematically and rigorously, however. One possible experiment is to explore whether using multiple AF2 models (there are five by default) would have an effect on pose accuracy, perhaps for selected examples such as NK1R, 5HT2A, and DRD1 to help build out the discussion further. Another possible weakness is that the authors focus only on GPCRs for reasons they state but make a good argument as to why the conclusions are likely to extend to other protein classes.

Context:

One of the most common and impactful uses of protein structures is in small molecule therapeutic chemical tool design. When no experimental structure is available, models are frequently used and such models include traditional template-based homology models and, more recently, AF2 models. AF2 is widely recognized as an inflection point in protein structure prediction due to the unprecedented accuracy of the protein structure models produced automatically. However, understanding whether this unprecedented accuracy translates to better small molecule ligand pose prediction has been an open question, and this study directly addresses the question in a systematic, rigorous approach.

---

## [Referee Report · Reviewer #2 (Public Review)]

While the question of 'are AlphaFold-predicted structures useful for drug design' has largely seen comparisons of AF versus experimental protein structures, this paper takes a less explored (but perhaps more practically important) angle of 'are AlphaFold-predicted structures any better than the previous generation of homology modeling tools' to the protein-ligand (rigid) docking problem. The conclusions of this work will be of largest interest to the audience less familiar with the precision required for successful rigid docking, while the expert crowd might find them obvious, yet a good summary of results previously shown in the literature. Further work, understanding the structural objectives/metrics that should be placed on future AlphaFold-like models for better pose prediction performance, would greatly expand the practicality of the observations made here.

The main conclusion of the paper, that structural accuracy (expressed as RMSD) of the protein model is not a good predictor of the accuracy the model will show in rigid docking protein-ligand pose prediction, is a good reminder of the well-appreciated need for high-quality side chain placements in docking. The expected phenomenon of AlphaFold predicting 'more apo-like structures' is often discussed in the field, and readers should be cautious about drawing conclusions from the rigid (rather than flexible, as in some previous works) docking done here.

The authors have very clearly communicated that the use of AlphaFold-generated structures in traditional docking might not be a good idea, and motivated that the time of a molecular designer might be better spent preparing a high-quality homology model. The visual presentation of the conclusions is very clear but might leave the reader wanting a more in-depth discussion of which structural elements of the AF models lead to bad docking outcomes. For example, Fig. 3 presents an example of a very accurate AlphaFold prediction leading to the ligand being docked completely outside of the binding pocket. Close inspection of the Figure suggests a clash of the ligand with the slightly displaced tryptophan residue in the AF model that might be to blame, as can be confirmed by comparison of the model and PDB structure by the reader themselves but has not been discussed by the authors. Only a few examples of the systems used are shown even visually, leaving the reader unable to study more interesting cases in depth without re-doing the work themselves.

The authors acknowledged that several recent studies exist in this space. They point out two advancements made in their work, worthy of further review. Similarly, it's important to evaluate the novelty of this work's claims vs previously available results, and the diversity of information made available to the reader.

"First, we use structural models generated without any use of known structures of the target protein. For machine learning methods, this requires ensuring that no structure of the target protein was used to train the method." This is done by limiting the scope of the work to GPCRs whose structures became available only after the training date of AlphaFold (April 30, 2018), as well as not using templates available after that date during prediction. The use of a time limit seems less preferable than the approach taken in Ref. 1 of discarding templates above a sequence identity cutoff. On the other hand, the 'ablation test' performed in Ref. 2 showed no loss in accuracy when no templates were used at all. Authors should discuss in more detail whether these modeling choices could change anything in their conclusions and why they made their choices compared to those in previous work.

"Second, we perform a systematic comparison that takes into account the variation between experimentally determined structures of the same protein when bound to different ligands." Cross-docking is indeed a more appropriate comparison than self-docking (as done in previous works), and the observation that the accuracy of AF models is similar to that between different holo structures of the same protein is interesting. Previous literature on cross-docking should however be discussed, and the well-known conclusions from it that small variations in side-chain positions, in otherwise highly similar structures, can lead to large changes in docked poses. It is important to realize that AlphaFold models are 'just another structure' - if previous literature is sufficient to show the sensitivity of rigid docking, doing it again on AF structures does not add to our understanding. Further, Ref. 3 might have already addressed the question of correlation between binding site RMSD and docking pose prediction accuracy - see e.g. Supplementary Figure 3 there (also Figure S15 in Ref. 2).

Further, the authors should discuss the commonly brought up problem of AlphaFold generating 'more apo-like structures' - are the models used here actually 'holo-like' because of the low RMSDs? (what RMSD differences are to be expected between apo and holo structures of these systems?) How are the volumes of the pockets affected? The position on this problem taken by previous works is worth mentioning - "much higher rmsd values are found when using the AF2 models (...), which reflect the difficulties in performing docking into apo-like structures" in Ref. 1 and "computational model structures were predicted without consideration of binding ligands and resulted in apo structures" in Ref. 2.

Because of this 'apo problem', Ref. 2 assumed that rigid docking (as done here) would not succeed and used flexible docking where "two sidechains at the binding site were set to be flexible". In fact, the reader of this new paper will be left to wonder if it is not simply presenting a subset of the results already seen in Ref. 2, where "the success ratios dropped significantly for them because misoriented sidechains prevented a ligand from docking (Figure S14)". While this conclusion is not made as clear in Ref. 2 as it is here, a comparison of Figures 4 and S14 there will lead the reader to the same conclusion, and more -- that flexible docking meaningfully improves the performance of AF models, and more so than homology models.

Finally, certain data analyses present in previous works but not here should be necessary to make this work more informative to the readers:

a) Consideration of multiple top poses, e.g., in Ref. 2, Figures 4 and S14 mentioned before, comparison of success rates in top 1 and top 3 docked poses add much context.

b) Notes on the structural features preventing successful docking, see e.g., in Ref. 1, Table 2 or in Ref. 4, Tables 2 and 4.

This work has the potential to become an important piece of the puzzle, if deeper insights into the reasons for AF model failures are drawn by the authors. These could include a discussion of the problematic structural elements (clashes of side chain with ligands, missing interactions/waters, etc.), potential solutions with some preliminary data (flexible docking, softening interactions, etc.), or proposals for metrics better than RMSD to score the soundness of pockets generated by AF for docking.

References:

1. Díaz-Rovira, A. M., Martín, H., Beuming, T., Díaz, L., Guallar, V., & Ray, S. S. (2023). Are Deep Learning Structural Models Sufficiently Accurate for Virtual Screening? Application of Docking Algorithms to AlphaFold2 Predicted Structures. Journal of Chemical Information and Modeling, 63(6), 1668-1674. https://doi.org/10.1021/acs.jcim.2c01270

2. Heo, L., & Feig, M. (2022). Multi-state modeling of G-protein coupled receptors at experimental accuracy. Proteins: Structure, Function, and Bioinformatics, 90(11), 1873-1885. https://doi.org/10.1002/prot.26382

3. Beuming, T., & Sherman, W. (2012). Current assessment of docking into GPCR crystal structures and homology models: Successes, challenges, and guidelines. Journal of Chemical Information and Modeling, 52(12), 3263-3277. https://doi.org/10.1021/ci300411b

4. Scardino, V., Di Filippo, J. I., & Cavasotto, C. (2022). How good are AlphaFold models for docking-based virtual screening? [Preprint]. Chemistry. https://doi.org/10.26434/chemrxiv-2022-sgj8c

---

## [Author Response]

We thank the reviewers and editor for their careful evaluation of our manuscript, and we appreciate their favorable assessment of our work. Below, we clarify a few points concerning the relationship between our study and previous studies evaluating ligand docking to protein models.

As reviewer 2 correctly notes, several previous assessments of AF2 models have simply excluded templates above a sequence identity cutoff when using AF2 to predict structures. Such AF2 predictions are still informed by all structures in the PDB before April 30, 2018, because these structures were used to train AF2—that is, to determine the tens of millions of parameters (“weights”) in the AF2 neural network. Machine learning methods nearly always perform better when evaluated on the data used to train them than when evaluated on other data. For this reason, we consider AF2 models only for proteins whose structures were not used to train AF2—that is, for proteins whose structures were not available in the PDB before April 30, 2018.

Previous papers (including Beuming and Sherman, 2012, https://doi.org/10.1021/ci300411b) have shown a clear correlation between the binding pocket RMSD of a protein model and pose prediction accuracy based on that model. Our main findings are unexpected in light of these previous reports: we find that AF2 models yield pose prediction accuracy similar to that of traditional homology models despite having much better binding pocket RMSDs, and that AF2 models yield substantially worse pose prediction accuracy than experimentally determined structures with different ligands bound despite having similar binding pocket RMSDs.

Reviewer 2 also correctly notes that previous papers have described AF2 models as “apo models,” because these models do not include coordinates for bound ligands. As noted by the AF2 developers (e.g., https://alphafold.ebi.ac.uk/faq), however, AF2 is designed to predict coordinates of protein atoms as they might appear in the PDB, and AF2 models are thus frequently consistent with structures in the presence of ligands even though those ligands are not included in the models. GPCR structures in the PDB, including those used to train AF2, nearly always contain a ligand in the orthosteric binding pocket. An AF2 model of a GPCR should thus not be viewed as an attempt to predict the GPCR’s structure in the unliganded (apo) state.

Finally, we did not apply flexible docking in this study because previous work has found that standard flexible docking protocols typically improve pose prediction performance only when given prior information on which amino acid residues to treat as flexible. For example, previous studies that performed successful flexible docking to AF2 models generally used prior knowledge of the ligand’s experimentally determined binding pose to identify the residues to treat as flexible.